# Electrospun Sulfonatocalix[4]arene Loaded Blended Nanofibers: Process Optimization and In Vitro Studies

**DOI:** 10.3390/pharmaceutics14091912

**Published:** 2022-09-09

**Authors:** Wan Khartini Wan Abdul Khodir, Shafida Abd Hamid, Mohd Reusmaazran Yusof, Iriczalli Cruz-Maya, Vincenzo Guarino

**Affiliations:** 1Department of Chemistry, Kulliyyah of Science, International Islamic University Malaysia Kuantan Campus, Bandar Indera Mahkota, Kuantan 25200, Pahang, Malaysia; 2SYNTOF, Kulliyyah of Science, International Islamic University Malaysia Kuantan Campus, Bandar Indera Mahkota, Kuantan 25200, Pahang, Malaysia; 3Agensi Nuklear Malaysia, Bangi, Kajang 43000, Selangor, Malaysia; 4Institute of Polymers, Composites and Biomaterials, National Research Council of Italy, Mostra d’Oltremare Pad.20, V.le J.F. Kennedy 54, 80125 Naples, Italy

**Keywords:** sulfonatocalixarene, electrospun fibers, polycaprolactone, gelatin, in vitro

## Abstract

In the past decade, electrospun nanofibers made of biodegradable polymers have been used for different biomedical applications due to their flexible features in terms of surface area to volume ratio, pores, and fiber size, as well as their highly tunable surface properties. Recently, interest is growing in the use of supramolecular structures in combination with electrospun nanofibers for the fabrication of bioactive platforms with improved in vitro responses, to be used for innovative therapeutic treatments. Herein, sulfonatocalix[4]arene (SCX4) was synthesized from *p*-*tert*-butyl-calix[4]arene and embedded in electrospun nanofibers made of polycaprolactone (PCL) and gelatin (GEL). The supramolecular structure of SCX4 and its efficient entrapment into electrospun fibers was confirmed by NMR spectroscopy and FTIR analysis, respectively. SEM analysis supported via image analysis enabled the investigation of the fiber morphology at the sub-micrometric scale, showing a drastic reduction in fiber diameters in the presence of SCX4: 267 ± 14 nm (without SCX) to 115 ± 5 nm (3% SCX4). Moreover, it was demonstrated that SCX4 significantly contributes to the hydrophilic properties of the fiber surface, as was confirmed by the reduction in contact angles from 54 ± 1.4° to 31 ± 5.5° as the SCX4 amount increased, while no effects on thermal stability were recognized, as was confirmed by TGA analyses. In vitro tests also confirmed that SCX4 is not cytotoxic, but plays a supporting role in L929 interactions, as was validated by the cell viability of PGC15% after 7 days, with respect to the control. These preliminary but promising data suggest their use for the fabrication of innovative platforms able to bind SCX4 to bioactive compounds and molecules for different therapeutic applications, from molecular recognition to controlled drug delivery.

## 1. Introduction

Over the past few decades, various types of macrocyclic compounds and their derivatives have been developed, including calix[n]arenes (CAs), crown ethers, cyclodextrins (CDs), cyclophanes, cucurbit[n]urils, and pillar[n]arenes [1,2,3], progressively gaining interest and popularity in the biomedical field. Among them, calixarenes, the third generation of macrocyclic hosts, are one of the most versatile classes of molecules, highly manipulated for a variety of applications that include biosensors, biomarker detection, drug discovery, and cell growth promotive materials [4,5]. Calix[n]arenes are cyclic basket-like macrocyclic compounds based on phenolic units (*n* = 4, 6, or 8) with defined upper and lower rims and a central annulus. The properties of calixarenes can be altered by modification of the rims resulting in a wide range of interesting functionalities for different applications, i.e., absorbents, catalytic supports, chemical sensors [6], biosensors [7], and drug carriers [8,9,10,11,12]. Only recently, an increasing number of works focusing on their biological response demonstrated that they are biologically friendly to the environment and exhibit good biocompatibility [13]. Owing to their promising biological activities such as antiviral, antibacterial, antifungal, and anticancer, functionalized calixarenes have received notable attention from the pharmaceutical/medicinal chemistry community [5]. Their host–guest complexation characteristic is particularly useful for designing stimuli-responsive supramolecular systems [14]. Most importantly, the cavities in the host molecules allow the accommodation of guest molecules as a function of the shape and size complementarity. Thus, high selectivity between the host and guest molecules provides strong dynamic interactions that allow for the synthesis of different supramolecular biomaterials with a high degree of structural complexity and programmable functions for different uses in the biomedical field. The strong reversible nature of their assemblies also offers the opportunity to mimic complex molecular recognition systems as they appear in nature (e.g., self-adapting to micro-environmental changes), for use in the development of bio-functional smart interfaces [15]. 

Recently, calix[n]arenes have been used to functionalize polymer-based biomaterials in different forms, such as nanoparticles [16,17,18], nanosponges [19,20], nanovalves [21], and nanofibers [8,22,23], enabling them to impart unique features, owing to their combination of chemical, physical, and morphological features [9,24]. In this context, the fabrication of supramolecular functionalized nanofibers is interesting because they successfully combine the unique properties of supramolecular interactions (i.e., they are non-covalent) with the advantages of nanofiber size scale, suited to the design of a wide range of nanostructured materials with highly tunable functionalities [25]. For example, Diao et al. [26] incorporated calix[8]arenes, ester-calix[8]arenes, and amide-calix[8]arenes into polyacrylonitrile (PAN) nanofibers with high catalytic activity and excellent stability, as well as convenient recycling. More recently, other studies have emphasized the unique chance to leverage supramolecular chemistry in combination with nanofibrous scaffolds for personalized therapeutic approaches in vitro. For this purpose, electrospinning is currently considered one of the best techniques to fabricate polymer nanofibers with supramolecular units, being based on a relatively noninvasive process that involves the use of electrical forces to manipulate polymer solutions or melt it into micro/nanofibers [27,28]. It enables the processing of a wide variety of synthetic biodegradable polymers such as Polycaprolactone (PCL), Poly(lactic-co-glycolic acid) (PLGA), Poly(lactic acid) (PLA), and Polyurethanes (PU) [29,30,31,32] at low cost and with high process efficiency. Due to the high versatility of the process, it easily enables the mixing of hydrophobic and hydrophilic materials—such as proteins, polysaccharides such as collagen, chitosan, gelatin, silk fibroin, fibrinogen, elastin, and keratin [31,32,33,34,35,36,37,38]—for biomedical use, thus overcoming the common limitations of several synthetic polymers in terms of cell affinity and poor cell adhesion capability [29]. For instance, PCL/gelatin nanofibers were extensively studied to reach a suitable compromise between good mechanical strength (due to the presence of PCL) and enhanced biocompatibility (due to the gelatin) for the fabrication of bicomponent fibers for use as bioinspired scaffolds for tissue engineering [29,33,39,40,41,42]. Other remarkable properties of electrospun nanofibers include high surface area to volume, high porosity, biodegradability, and flexibility for chemical/physical functionalization as variously described in literature [29,40].

In this context, the integration of supramolecular species may be suitable to bind specific ligands to improve cell adhesion and migration, and to regulate cellular proliferation and functions to promote tissue regeneration in vitro as a native extracellular matrix does in vivo [33,43]. However, one of the major problems of using calix[n]arenes in biomedical applications is their total insolubility in aqueous solution. In order to overcome this problem, recent studies are exploring the incorporation of water-soluble moieties such as sulfonate groups at the phenyl ring as a strategy to provide a strong hydrophilic upper rim and hydrophobic inner cavity [44,45]. To date, only few reports are currently available on the functionalization of nanofiber scaffolds using SCX4 for biomedical applications [45,46].

In this work, we proposed to investigate the use of water-soluble SCX4 in combination with PCL/Gelatin nanofibers as a supramolecular hybrid system with improved cell adhesion and surface properties. Briefly, the synthesis of water-soluble SCX4 was optimized to functionalize the surfaces of the PCL/gelatin nanofibers, fabricated via electrospinning. It was verified that this functionalization may contribute to a reduction in cytotoxicity, thus improving the biocompatibility of the nanofibers due to the efficient host–guest interactions of SCX4. The chemical and physical properties of the supramolecular-functionalized nanofibers were investigated using NMR, FTIR, and TGA analyses. Then, hydrophilicity and cell interaction were evaluated via SEM, water contact angle, and in vitro studies by the use of a fibroblast cells taken from the subcutaneous connective tissue of mice (L929).

## 2. Materials and Methods

### 2.1. Materials

*p*-*tert*-Butylcalix[4]arene was obtained from Acros (Morris Plains, NJ, USA). Polycaprolactone (PCL; M_w_ = 45,000), gelatin from bovine skin (gel. strength 225 bloom, Type B), and 1,1,1,3,3,3-Hexafluoro-2-propanol (HFIP) were purchased from Sigma-Aldrich. For the cell biocompatibility response to PCL and PCL/water soluble calixarene, mouse fibroblasts cells (L929) were used. L929 cells were cultured in a 75 cm^2^ cell culture flask in Dulbecco’s Modified Eagle Medium (DMEM, Sigma, Milan, Italy), supplemented with 10% Fetal Bovine Serum (FBS, Sigma Aldrich, Milan, Italy), antibiotic solution (streptomycin 100 µg/mL, penicillin 100 U/mL, and fungizone 0.3 µg/mL, Sigma), and 2 mM L-glutamine (Sigma), incubated at 37 °C in a 100% humidified atmosphere with 5% CO_2_ and 95% air. All chemicals or reagents were used as received without further purification.

### 2.2. Synthesis of p-Sulfonatocalix[4]arene

Calix[4]arene was synthesized following the published method [47]. *p*-*tert*-Butylcalix[4]arene (25 g, 38.5 mmol), phenol (20 g, 0.21 mol), and toluene (250 mL) were cooled to 10 °C, and anhydrous AlCl_3_ (27.5 g, 0.21 mol) was added to the solution. The reaction mixture was stirred overnight at room temperature. The reaction mixture was poured onto 2 M HCl (160 mL), and stirred vigorously for 30 min, after which the organic phase was separated, and the solvent was evaporated. CHCl_3_ (110 mL) was added to this suspension, which was refluxed for 30 min, and then MeOH (160 mL) was added, which triggered precipitation. The mixture was filtered after 4 h and dried in a vacuum overnight at 50 °C to obtain a fine white powder (98%); mp 340.2 °C; IR (KBr) *ν*_OH_ 3152 cm^−1^. ^1^H-NMR (CDCl_3_): δ 3.48 and 4.26 (ring CH_2_, br, 8H), 6.72 (Ar H, m, 4H), 7.05 (Ar H, m, 8H), 10.19 (-OH, s, 4H). ^13^C-NMR (CDCl_3_): δ 31.76 (ring CH_2_), 122.3 (s), 125.9 (s), 129.0 (s), 148.8 (C-OH).

The Calix[4]arene was then converted into *p*-sulfonatocalix[4]arene using Shinkai’s method [48,49] with slight modification. The Calix[4]arene (13 g, 30.6 mmol) was mixed with 130 mL of concentrated H_2_SO_4_, and the solution was heated at 80 °C for 3 h. The reaction was completed when no water-insoluble material was detected. After cooling, the precipitate was recovered by filtration. The precipitate was dissolved in 50 mL of water and neutralized with brine (260 mL) to obtain a fine white powder (90%); mp > 400 °C; ^1^H-NMR (D_2_O): δ 3.32 and 4.12 (ring CH_2_, s, 8H), 7.68 (Ar H, s, 8H). ^13^C-NMR (D_2_O): δ 30.5 (ring CH_2_), 126.5 (s), 128.1 (s), 135.7 (C-SO_3_^−^), 151.6 (C-OH).

### 2.3. Preparation of Electrospun Nanofibers

10% *w*/*v* of PCL/gelatin (1:1) was dissolved in 1,1,1,3,3,3-Hexafluoro-2-propanol (HFIP) solvent at room temperature for 24–48 h. Then, the PCL/gelatin solution (coded as PG) was mixed with selected amounts of SCX4—3, 5, 10, and 15 mg for 10 mL of solution (coded as PGC-3, PCG-5, PGC-10, and PGC-15, respectively)—and stirred for 24 h until it formed a homogenous and clear solution. Preliminary experimental trials were performed to evaluate the molecular solubility at different SCX4 concentrations, identifying 15 mg/mL SCX4 in polymer solution as the saturation limit (at this concentration, the solution does not become completely clear after 24 h). The solutions were processed using a built-up electrospinning machine equipped with a high voltage (Gamma High Voltage, Ormond Beach, FL, USA) working at 13 kV and a syringe pump (KD Scientific, Holliston, MA, USA) with flow rate set to 0.8 mL/h. The distance between the ground collector and the needle (20 G) was fixed to 100 mm. Electrospun fibers were fabricated under standard environmental conditions—i.e., a temperature of 28 °C and relative humidity of 50 ± 5%—and, after the deposition, dried at room temperature to remove any residual solvents. All the samples were stored in a desiccator for further characterization.

### 2.4. Chemical and Physical Characterization

#### 2.4.1. ^1^H and ^13^C Nuclear Magnetic Resonance Spectroscopy

^1^H and ^13^C-NMR spectra of the polymeric materials dissolved in deuterated water were recorded on a Bruker spectrometer at 400 MHz and 100 MHz, respectively. Tetramethylsilane (TMS) was used as the internal reference. All chemical shifts were reported in parts per million (ppm).

#### 2.4.2. Scanning Electron Microscopy (SEM)

The surface morphologies of PG and PGC 3–15 nanofibers were obtained by using scanning electron microscopy (JEOL 6360 LA, Tokyo, Japan). The samples were sputter-coated with gold nano-powder. The average fiber diameter distributions were estimated using Image J^®^ software (upgrade version 1.53/2022) (National Institute of Mental Health, Bethesda, MD, USA) using 100 random measurements based on selected SEM images.

#### 2.4.3. FTIR Spectroscopy

SCX4 chemical interactions were analyzed using Attenuated Total Reflectance-Fourier Transform Infrared Spectroscopy (Perkin Elmer, Waltham, MA, USA). The FTIR spectra were measured in the spectral range of 600 to 4000 cm^−1^, performed at 16 scans per sample.

#### 2.4.4. Water Contact Angle

Water contact angle measurements were conducted to evaluate the hydrophilicity of the nanofiber surfaces by using a water contact angle apparatus (WCA, One Attention Theta, TL100, Biolin Scientific, Espoo, Finland). Two μL of distilled water was dropped on the nanofibrous surface. An average of 3 measurements was considered to calculate the final contact angle value.

#### 2.4.5. Thermogravimetric Analysis

The thermal properties were analyzed by thermogravimetric analysis (TGA-50, Shimadzu, Tokyo, Japan). An amount of approximately 3 mg was sealed in a nonhermetic aluminum pan under a nitrogen atmosphere. Samples were scanned from 30 °C up to 600 °C at 10 °C/min, under gaseous nitrogen flow. Weight loss against temperature was recorded in the thermogram.

### 2.5. In Vitro Studies

L929 cells were seeded onto the electrospun fibers at 2 × 10^4^ cells/well and were incubated under standard conditions at 12 and 24 h. After this period, cell adhesion was evaluated by toluidine blue (TB, Sigma). Briefly, electrospun fibers were rinsed with phosphate buffer solution (PBS) three times to remove the non-adherent cells. Then, cells were fixed with 4% paraformaldehyde and incubated with 0.1% of TB for 3 h. For adhesion measurements, the dye was extracted with 0.1% of sodium dodecyl sulfate (SDS). The optical absorption was quantified by spectrophotometry (Perkin-Elmer, Waltham, MA, USA) at 600 nm with a plate reader. A tissue culture plate (TCP) was used as a control. Experiments were conducted in triplicate. The cell viability of the L929 cells was evaluated via XTT assay (Roche, Milan Italy) after they were seeded onto electrospun fibers at 1 × 10^4^ cells/well during 1, 3, 7, and 14 days. The XTT assay was analyzed based on the reduction of tetrazolium salt XTT to a soluble formazan salt by viable cells in the presence of an electron coupling reagent. The activated XTT solution was prepared, added to the samples, and incubated for 4 h. The supernatant was removed and placed in a plate reader to measure the absorbance of formazan at 450 nm. The experiments were conducted in triplicate and repeated three times. The results were presented as mean ± standard deviation.

## 3. Results and Discussion

(a)Synthesis of p-sulfonatocalix[4]arene

The synthesis started with the dealkylation of *p-tert*-butylcalix[4]arene (Figure 1) in the presence of phenol, toluene, and AlCl_3_ at room temperature to give a 98% yield of product. The ^1^H-NMR spectrum of compound **1** shows the characteristic peaks for the methylene bridge protons at δ 3.48 and 4.26 ppm, indicating its existence as a cone conformation (as in Figure 2). The ^13^C-NMR peak at δ 31.7 ppm also shows the presence of methylene bridge carbon. The disappearance of the proton peaks of the *p*-*tert*-butyl group is evidenced by the ^1^H-NMR spectrum and by the presence of two peaks at δ 6.72 and 7.05 ppm, along with the ^13^C-NMR peaks at δ 122.3, 125.9, and 129.0 ppm.

Calix[4]arene (**1**) was then used as the next starting material for *p*-sulfonatocalix[4]arene (**2**) by treating the compound with concentrated H_2_SO_4_ to produce the desired product in 90% yield. The proton NMR chemical shift at δ 7.47 (Ar H, m, 8H) shows the para-substituted benzene rings, while the carbon attached to the sulfonate peak is indicated at δ 135.7 ppm. Compound **2** also has a cone conformation as indicated by the ^1^HNMR spectrum, with the characteristic methylene bridge proton peaks at δ 3.32 and 4.12 ppm (Figure 2). This data is in agreement with the ^13^C-NMR methylene bridge peak at δ 30.5 ppm.

(b)Morphological studies

SEM images of PGC, PGC-3, PGC-5, PGC-10, and PGC-15 nanofibers are shown in Figure 3. Non-aligned and bead-free smooth fibers were achieved by a fine optimization of the process parameters. Pristine PG nanofibers (Figure 3a) show an average fiber diameter of about 267 ± 14 nm. The addition of 3% SCX4 (PGC-3) (Figure 3b) significantly reduces the characteristic sizes by 40%, with an average fiber diameter equal to 115 ± 5 nm. It is noteworthy that only slight differences without remarkable trends can be recognized as the SCX4 amount increases: PGC-5, PGC-10, and PGC-15 show average fiber diameters equal to 134 ± 9 nm, 159 ± 9 nm, and 152 ± 3nm, respectively. In the case of PGC-10 and PGC-15, some beads, nodes, and fiber defects may be preferentially recognized (Figure 3d,e). Independently of the relative amount, the presence of SCX4 does not alter the fibrillary architecture of the scaffolds. Nevertheless, there was an evident downsize of fiber mesh when compared with the controls (i.e., nanofibers without SCX4). The decrease in fiber diameter can be attributed to the influence of SCX4 on polymeric solution properties that contribute to the production of more elongated, split off, and thinner nanofibers [28,50]. Indeed, the supramolecular structure of SCX4 affects solvent interactions under the application of the electric field, providing a more extended interface that promotes efficient evaporation mechanisms that facilitate the formation of thinner fibers compared with the control. However, the heterogeneous dispersion of SCX4 in solution also promotes the formation of a broad distribution of fiber diameters, inducing the formation of the thinnest fibers in the case of the intermediate SCX4 concentration. Nevertheless, these peculiar structural characteristics are also attributed to the type of solvent used, SCX4 concentration, and their intermolecular interaction, as reported in a previous work by Celebioglu et al., which focused on electrospun M-β-CD (methyl-beta-cyclodextrin) nanofibers [1]. In this work, the use of HFIP as a solvent was highly recommended for processing supramolecular structures into nanofibers, due to the high polarity and good dielectric constant, which makes the electrospinning much easier [51].

(c)FTIR analysis

The FTIR spectra of PG, SCX4, PGC-3, PGC-5, PGC-10, and PGC15 are shown in Figure 4. The bands observed at 3304 cm^−1^ (OH broadening), 2944 cm^−1^ (asymmetric CH_2_ stretching), 2866 cm^−1^ (symmetric CH_2_ stretching), 1726 cm^−1^ (carbonyl stretching), 1640 cm^−1^ (amide I), 1542 cm^−1^ (amide II), and 1177 cm^−1^ (asymmetric C-O-C stretching) correspond to the functional groups in both PCL and gelatin (Figure 4a) [32,52]. SCX4 (Figure 4b) is identified by strong bands at 2955 cm^−1^ (CH stretching), 1628 cm^−1^ (C=C bond), 1459 cm^−1^ (asymmetric C=C stretching), 1286 cm^−1^, and 1238 cm^−1^ (C-O bond stretching vibration). The S-O stretching vibration is indicated by the peaks at 1184 cm^−1^, 1118 cm^−1^, and 1042 cm^−1^. The bands observed at 3430 cm^−1^ and 3173 cm^−1^ indicate the broadening of the O-H band due to the polar SO_3_ groups participating in the H-bonding with water molecules in the calixarene cavity.

The oscillations of the OH groups along the lower rim of the calixarene molecule are shown by the peak at 3188 cm^−1^ [53]. We attribute this to the formation of strong hydrogen bonds. The SCX4 functionalized PG nanofibers at various amounts (Figure 4c–f) display the absorption bands at around 1047 cm^−1^, 1176 cm^−1^, and 1184 cm^−1^, corresponding to the S-O stretching vibration, while the C-C absorption band is shown at 1459 cm^−1^. A broadening of the O-H bands at 3173–3430 cm^−1^ is due to the O-H stretching of SCX4 and gelatin. When the SCX4 amount increases (i.e., PGC-10, PGC-15), the peaks at 3296 cm^−1^ (O-H broadening) and 1176 cm^−1^ (S-O stretching vibration) related to the OH and SO_3_ groups showed a slight reduction in intensity. This effect is probably due to the increased hydrogen bonds, also corroborated by the incipient saturation of SCX4 in the solution [54].

(d)Water contact angle

The water contact angle measurements of the nanofibers were carried out, and the results are summarized in Figure 5. The water contact angle of pure PCL nanofibers is 118 ± 6°, which confirms its hydrophobic nature, as reported in a previous study [33]. The PG nanofibers blended at a 1:1 ratio gave a water contact angle of 54 ± 1.4°, demonstrating higher wettability than the PCL nanofibers. The results indicate that the PG scaffold is hydrophilic, as it can form a hydrogen bond with the water, while pure PCL is hydrophobic due to the presence of CH_2_ groups in the main chain of PCL [53,55]. Because the SCX4 is functionalized with PG nanofibers, the water contact angles for PGC-3, PGC-5, PGC-10, and PGC-15 nanofibers were reduced to 49 ± 1.4°, 34 ± 2.8°, 31 ± 1.5°, and 31 ± 5.5°, respectively. The higher the SCX4 content, the more hydrophilic the scaffolds, leading to a higher wetting tendency. The presence of OH and SO_3_H moieties in SCX4 clearly contributed to the more hydrophilic nature of the surface of the scaffolds, due to the effect of SCX4 interaction. This property is very important as it influences cellular adhesion, proliferation, and differentiation [56].

(e)Thermogravimetric analysis

The thermal stability of the SCX4, PG, and PGC nanofibers was determined by thermogravimetric analysis (Figure 6). SCX4, which contains polar sulfonate groups, attracts more water molecules, and the compound showed mass loss at two stages. The first stage of weight loss at about 50 °C corresponds to the loss of physically adsorbed water, and the second stage occurred at 408–535 °C (Figure 6a). The maximum rate of mass loss at 460 °C (Figure 6b) is associated with the separation of the sulfonate groups and the destruction of the calixarene structure [52]. By the elaboration of TGA curves for PG and PGC nanofibers, thermal degradation was recognized by three stage mechanism (Table 1): 320–333 °C, 390–396 °C, and 530–560 °C. The significant decomposition of PG and PGC nanofibers started at around 320 °C and was due to the rupture of the ester chains, which led to the formation of H_2_O, CO_2,_ and 5-hexenoic acid from PCL, and was also associated with protein degradation [26,57]. The degradation continued at 532–560 °C, with almost 70% weight loss, which resulted from the rupture and decomposition of the PCL, gelatin, and SCX4. The addition of SCX4 into the PG matrix improved the stability of the nanofiber scaffolds as a higher temperature was then required to start the decomposition process. As the SCX4 content was increased, the weight loss temperature also increased, indicating the consistent stabilizing effect of calixarene.

(f)Biocompatibility studies

An in vitro study using an L929 cell line was performed to investigate the biocompatibility of SCX4 functionalized nanofibers. L929 is a fibroblast cell line from the subcutaneous connective tissue of mice that is considered one of the most reliable cell models to investigate in vitro cytotoxicity. Moreover, after several passages, they transform into cells that show some features of cancerous cells, thus resulting an interesting in vitro model to potentially investigate in the future, i.e., the release of chemotherapeutic drugs via calixarenes.

Firstly, cell adhesion was evaluated after 12 and 24 h (Figure 7). Based on the cell culture plate, the cell adhesion was over 50% after 12 h, although no significant differences between the samples were detected. However, after 24 h, an improvement in the cell adhesion was observed, showing significant differences between PG and all the other groups with calixarene content, except PGC-3, which showed similar behavior.

For cell viability, an XTT assay was employed after 1, 3, 7, and 14 days in cell culture (Figure 8). The PG and PGC-3 showed better cell viability than cell adhesion after 1 day. However, no significant differences were detected.

After 3 days in cell culture, the viability of L929 cells was increased in all samples, but PGC-3 displayed metabolic arrest cell activity. A similar trend was observed after 7 days, with a recovery in cell viability for PGC-3 as well. Furthermore, at this time point, the PGC-10 and PGC-15 samples presented an improvement in cell viability compared with PG, thus, no significant differences were detected between these groups. After 14 days, cells seeded onto the PG, PGC-10, and PGC-15 samples showed good metabolic activity, although lower when compared with the results observed after 7 days, which may be a result of the high confluence reached by the electrospun nanofibers at day 7. In other words, these results confirm the interaction of SCX4 with gelatin macromolecules. As the concentration of SCX4 increases, a more effective interaction and a high capability of the fibers to retain gelatin macromolecules is achieved, thus demonstrating the cell viability up to 7 days.

## 4. Conclusions

In this work, we outlined the preparation of electrospun PCL/Gelatin fibers functionalized with SCX4 at different concentrations. Macrocyclic compounds were synthesized and characterized in order to be combined with polymeric nanofibers for biomedical use. The use of electrospinning technology allowed for an accurate control of the characteristic sizes (i.e., fiber size and distribution) of the scaffold. The SEM results showed that the incorporation of SCX-4 contributed to a drastic reduction in the average fiber diameter (as much as 40%) compared with those of PG. Moreover, the addition of SCX4 influenced the final morphology of the nanofibers thanks to the interactions of their functional groups with PCL/gelatin fibrous networks, as confirmed by the ATR-FTIR studies. In this context, the presence of supramolecular materials with sizes in the nanometer range also enabled the significant enhancement of fiber surface hydrophilicity, and thermal stability, thus improving the biocompatibility in terms of cell adhesion and viability. The results, even if preliminary, suggest a potential use of SCX4 as an amphiphilic carrier to bind selected bioactive molecules or drugs for the fabrication of nanofibrous platforms with multiple functionalities, from ion detection to a programmed delivery of pharmaceutical species. Recent works highlight the potential use of calixarenes as carriers of antiviral, antibacterial, anti-inflammatory, antifungal, anti-tubercular, and anticancer agents, with optimal molecular bioavailability and efficacy, and limited side effects. Thus, there is an urgent need for the implementation of customized experimental setups to design fibrous platforms with even more complex architectures, able to confine supramolecular species in specific compartments of the fibers (core shell, Ja-nus, etc.) [58,59,60]. This will pave the way for the fabrication of much more innovative diagnostic and/or therapeutic systems with releasing, targeting, and detection capabilities, and towards new approaches to the care of healthy or cancerous tissues.

## Figures and Tables

**Figure 1 pharmaceutics-14-01912-f001:**
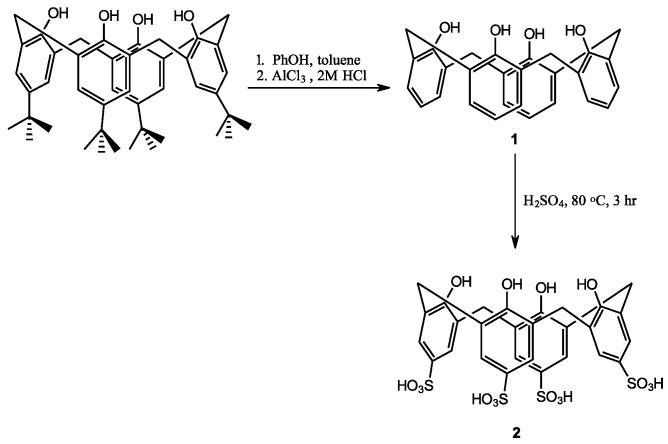
Reaction scheme of *p*-sulfonatocalix[4]arene synthesis.

**Figure 2 pharmaceutics-14-01912-f002:**
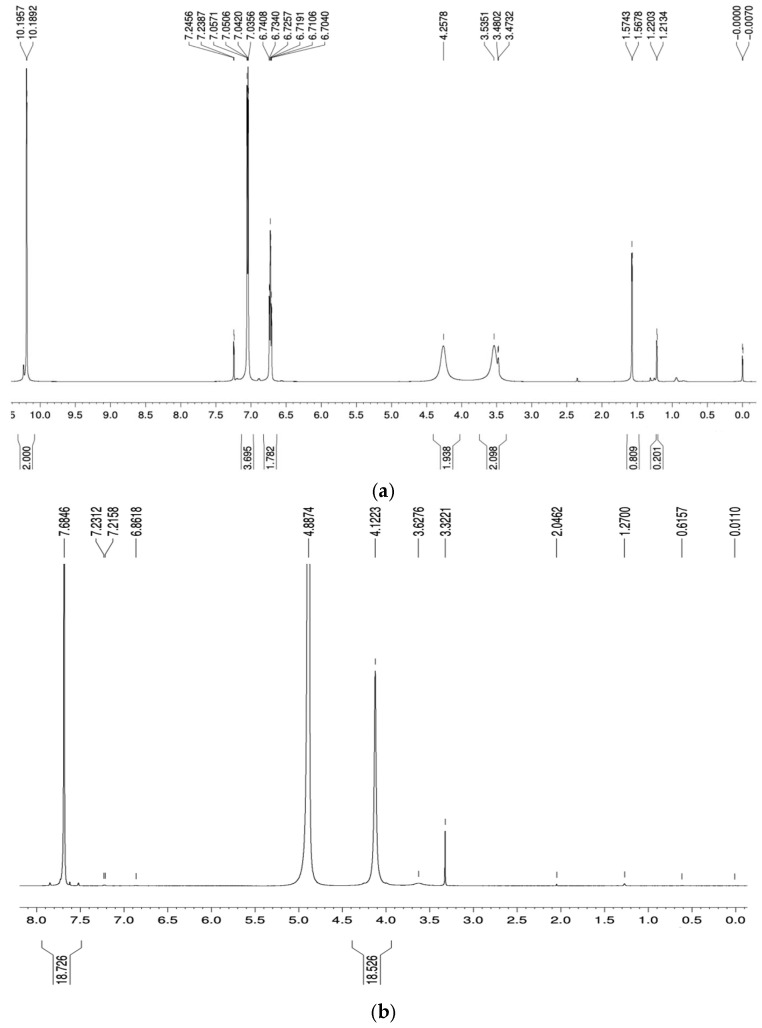
^1^H-NMR spectrum of (**a**) calix[4]arene and (**b**) SCX4.

**Figure 3 pharmaceutics-14-01912-f003:**
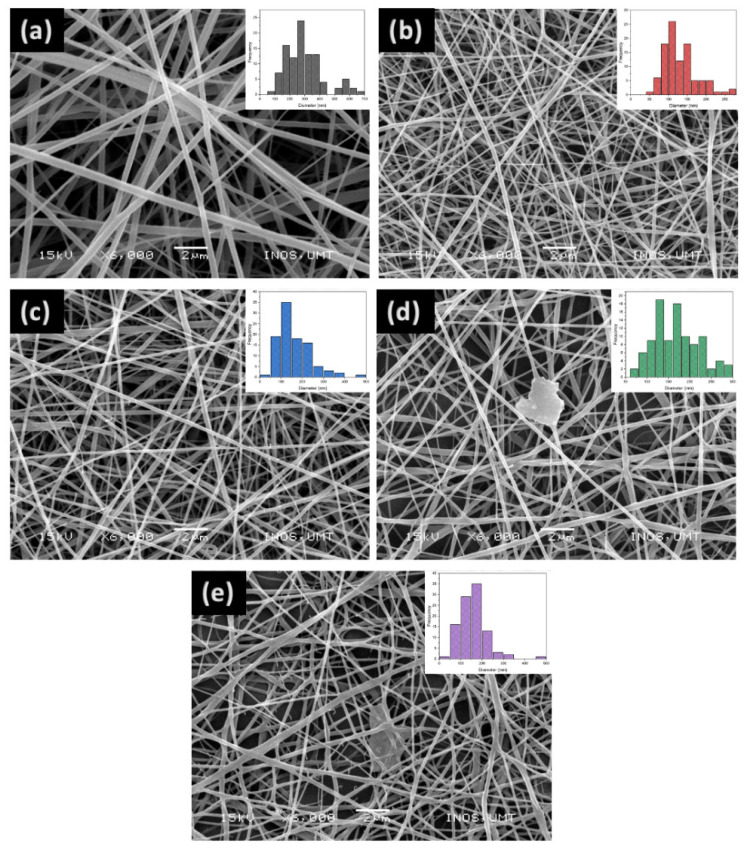
SEM images of electrospun nanofibers from (**a**) PGC, (**b**) PGC-3, (**c**) PGC-5, (**d**) PGC-10 and (**e**) PGC-15 nanofibers at 6000 magnifications. The insets show the fiber diameter distributions. Scale bar: 2 µm.

**Figure 4 pharmaceutics-14-01912-f004:**
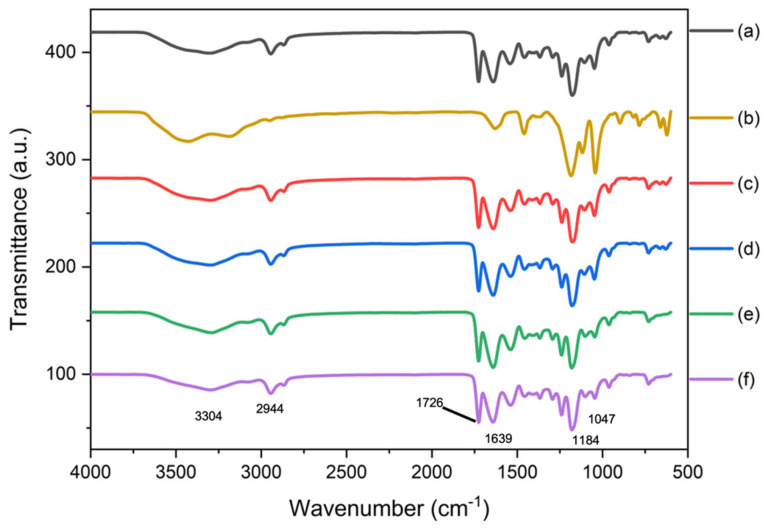
FTIR spectrum of (a) PG, (b) SCX4, (c) PGC-3, (d) PGC-5, (e) PGC-10, and (f) PGC-15.

**Figure 5 pharmaceutics-14-01912-f005:**
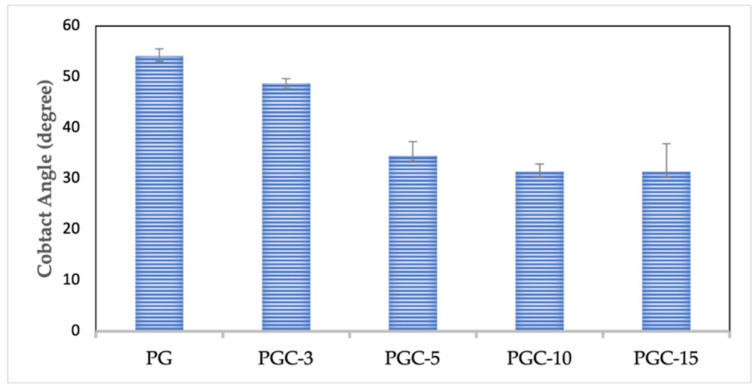
Water contact angle measurements of PG and PGC nanofibers.

**Figure 6 pharmaceutics-14-01912-f006:**
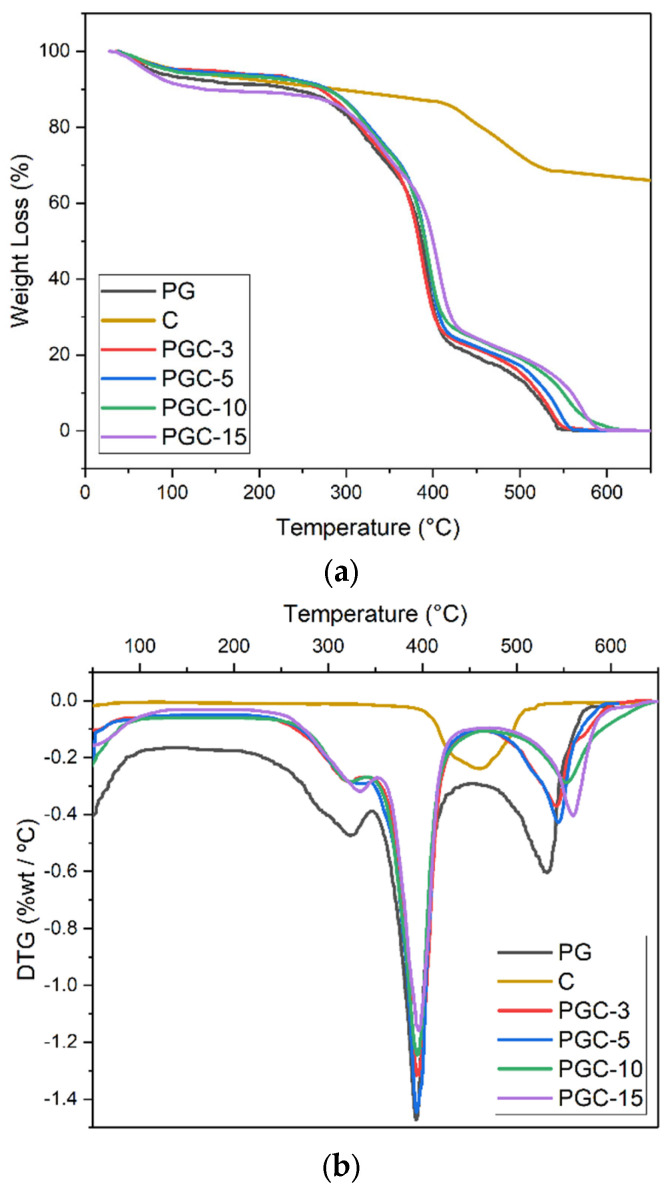
Weight loss of PG and PGC nanofibers (**a**) TGA, (**b**) DT.

**Figure 7 pharmaceutics-14-01912-f007:**
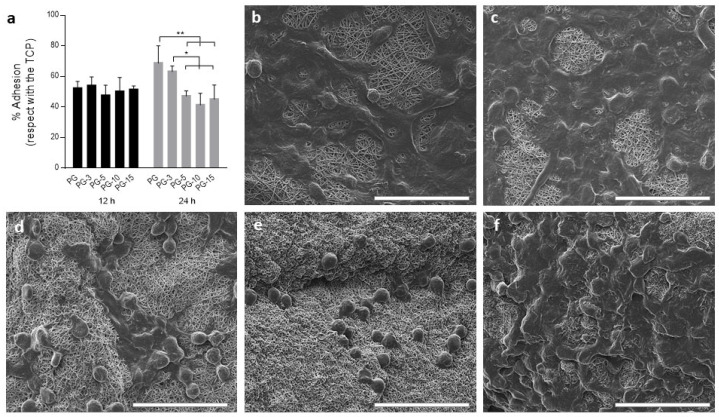
Cell adhesion of L929 cells. (**a**) Quantitative evaluation after 12 and 24 h in in vitro culture calculated as percentage of adhered cells with respect to the tissue culture plate (TCP, 100%). Results are presented as mean ± standard derivation. * *p* < 0.05, ** *p* < 0.01; (**b**–**f**) qualitative evaluation of cell adhesion via SEM on PG, PG-3, PG-5, PG-10, and PG-15, respectively. Scale bar: 50 µm.

**Figure 8 pharmaceutics-14-01912-f008:**
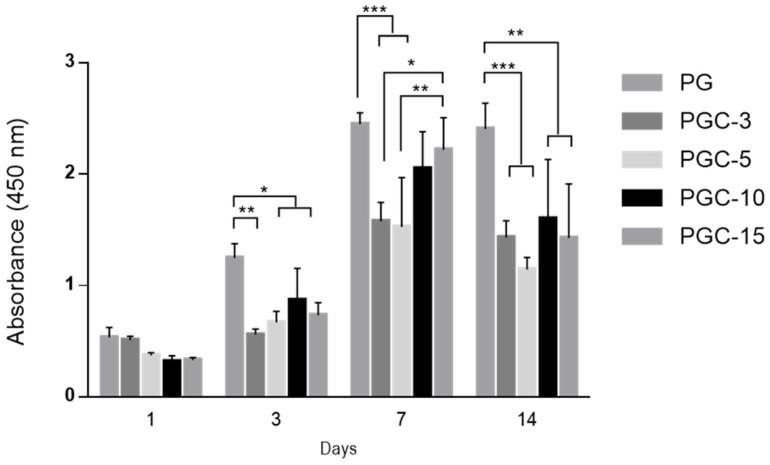
Cell viability of L929 cells on different samples after 1, 3, 7, and 14 days. Results are presented as mean ± standard derivation * *p* < 0.05, ** *p* < 0.01, *** *p* < 0.001.

**Table 1 pharmaceutics-14-01912-t001:** TGA results of PG, SCX4, and PGC.

Samples	Main Region of Decomposition (%)	Percentage of Mass Decomposition (%)	T_max1_ (°C)	T_max2_ (°C)	T_max1_ (°C)
PG	90–500	77.00	323.24	393.52	532.21
SCX4	420–500	21.90	50	461.28	-
PGC-3	90–500	75.67	321.91	394.83	542.22
PGC-5	90–500	73.96	336.42	393.98	544.60
PGC-10	90–500	70.00	323.43	395.09	552.91
PGC-15	90–500	67.61	333.38	396.15	560.25

## Data Availability

Not applicable.

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
