# Peer review of "Electrospun Sulfonatocalix[4]arene Loaded Blended Nanofibers: Process Optimization and In Vitro Studies"

_pharmaceutics, 2022, doi:10.3390/pharmaceutics14091912_

Round 1
Reviewer 1 Report
Authors have fabricated novel sulfonate Calix[4]Arene functionalized bioactive nanofibers and investigated their structural as well as biological properties. The study is rigorous with an organized approach to demonstrate the strength of their materials for biomedical application. However, some flaws have been observed and need to be corrected in order to elevate the scientific value of their study. Therefore, I recommend the following minor revision to be addressed to be suitable for publication in pharmaceutics.
To address,
1) To strengthen your introduction about electrospinning, consider citing the following papers:
(ACS Appl. Mater. Interfaces 2017, 9, 19, 16381–16396. 10.1021/acsami.7b00970) (Journal of Drug Delivery Science and Technology 57 (2020) 101604. 10.1016/j.jddst.2020.101604)
2) Which application do you think is particularly fitted within the biomedical field for your system?
3) You mention a change of fibers morphology and dimension, what is the origin of such an observation and how does it impact the polymer solution?
4) On figure 4, please consider adding the most important peaks signal to highlight key points for the reader understanding. Also, for the contact angle measurement, if you have photograph, please provide them.
5) Correct the English here and there.
Author Response
Authors have fabricated novel sulfonate Calix[4]Arene functionalized bioactive nanofibers and investigated their structural as well as biological properties. The study is rigorous with an organized approach to demonstrate the strength of their materials for biomedical application. However, some flaws have been observed and need to be corrected in order to elevate the scientific value of their study. Therefore, I recommend the following minor revision to be addressed to be suitable for publication in pharmaceutics.
1) To strengthen your introduction about electrospinning, consider citing the following papers: (ACS Appl. Mater. Interfaces 2017, 9, 19, 16381–16396. 10.1021/acsami.7b00970) (Journal of Drug Delivery Science and Technology 57 (2020) 101604. 10.1016/j.jddst.2020.101604)
Thank you for your suggestion. The second paper has been cited into the introduction and included as ref 36.
2) Which application do you think is particularly fitted within the biomedical field for your system?
Thank you for your comment. Due to their peculiar chemical–physical properties, calixarenes can be easily functionalized by different kind of molecules demonstrating their potential in the biomedical application. Calixarene have been extensively studied for their beneficial effects on human health especially as antiviral, antibacterial, anti-inflammatory, antifungal, antitubercular and anticancer agent that is useful in biomaterials field. Calixarene is widely known as a host for bioactive guest molecules/drug and therefore capable of releasing drugs in a controlled manner. Recently, they have been successfully used as efficient carrier of anticancer drugs, due to the ability to improve molecular bioavailability and efficacy, reducing their side effects and increasing their affinity towards the target. Moreover, Calixarene was considered an important receptor for molecular recognition suitable for bio-sensing application. Some sentences have been included in the introduction and conclusion section to better underline the applicative goals of SCX4 and functionalized devices.
3) You mention a change of fibers morphology and dimension, what is the origin of such an observation and how does it impact the polymer solution?
Thank you for your question. Some sentences have been included in the results and discussion section to better discuss the reported data as follows:” The decrease in fiber diameter can be attributed to the influence of SCX4 on polymeric solution properties, that contributes to produce more elongated, split off, and thinner nano-fibers [28,50]. Indeed, the supramolecular structure of SCX4 affects solvent interactions under the application of the electric field, providing more extended interfaced that promote efficient evaporation mechanisms that concur to the formation of thinner fibres respect to the control. However, the heterogeneous dispersion of calixerene in solution also promotes the formation of a broad distribution of fibre diameters, inducing the formation of thinnest fibres in the case of intermediate SCX4 concentration”.
4) On figure 4, please consider adding the most important peaks signal to highlight key points for the reader understanding. Also, for the contact angle measurement, if you have photograph, please provide them.
Figure 4 has been revised, by labelling most relevant peaks. As for the contact angle measurement, unfortunately, droplet images were not saved during the analysis.
5) Correct the English here and there.
Thank you, the manuscript has been read carefully and revised to remove misprints and mispellings. All the changes have been yellow marked into the text.
Reviewer 2 Report
With Sulfonato Calix[4]arene (SCX4) as a functional ingredient, a blended nanofibers containing polycaprolactone (PCL) and gelatin (GEL) as filament-forming matrices were prepared using a blending electrospinning process. The results successfully demonstrated the desired functional performances. The contents are interesting and falls well within the scope of PHARMACEUTICS. I recommend its acceptance for publication after the following issues are addressed.
The title can be further condensed, such as “Electrospun Sulfonato Calix[4]Arene loaded blended nanofibers”, I see no details about the process optimization of electrospinning, such as the matched applied voltage, the fluid flow rate, the fiber collection distance, and the reasonable loading of SCX4.
Several sentences in the INTRODUCTION about the developments of electrospinning should be included for provoking the readers’ interests (e.g. SCX is loaded in the core of core-shell nanofibers or one side of Janus nanofibers), and in turn the impact of your article after publication, e.g. electrospinning is fast developing from the single-fluid bledning processes (https://doi.org/10.3390/pharmaceutics14071494;Pharmaceutics, 4(3),484; Pharmaceutics, 4(4),777) to coaxial (DOI10.3390/pharmaceutics14071442; ), triaxial (https://wires.onlinelibrary.wiley.com/doi/full/10.1002/wnan.1772), side-by-side (https://doi.org/10.1016/j.bioadv.2022.212795), tri-layer side-by-side processes (Doi: 10.1007/s42114-022-00478-3) for creating nanostructures. However, these structures are similar as the monolithic nanofibers that active ingredients must be loaded into them to endow the designed functional performances.
What is the purity of HFIP? How about the saturated solubility of SCX4 in it?
The scale bars in Figure 3 please be enlarged.
Line 291, Water contact angle.
The references’ formats please be unified, and the most recent three years’ references can be more, a common standard is 25%, particularly for a fast developing region. PHARMACEUTICS has many excellent publications about electrospinning. The authors are suggested to discuss them with your job.
Author Response
Reviewer 2:
With Sulfonato Calix[4]arene (SCX4) as a functional ingredient, a blended nanofibers containing polycaprolactone (PCL) and gelatin (GEL) as filament-forming matrices were prepared using a blending electrospinning process. The results successfully demonstrated the desired functional performances. The contents are interesting and falls well within the scope of PHARMACEUTICS. I recommend its acceptance for publication after the following issues are addressed.
1) The title can be further condensed, such as “Electrospun Sulfonato Calix[4]Arene loaded blended nanofibers”, I see no details about the process optimization of electrospinning, such as the matched applied voltage, the fluid flow rate, the fiber collection distance, and the reasonable loading of SCX4.
Thank you for your suggestion, the title has been modified accordingly. As for the description of the preparation, it has been revised as follows: ”10% w/v of PCL/gelatin (1:1) was dissolved in 1,1,1,3,3,3-Hexafluoro-2-propanol (HFIP) solvent at room temperature for 24- 48 hours. Then, PCL/gelatin solution (coded as PG) was mixed with selected amounts of SCX4 – 3, 5, 10, and 15 mg for each mL of solution (coded as PGC-3, PCG-5, PGC-10, and PGC-15, respectively) and stirred for 24 hours until to form a homogenous and clear solution. Preliminary experimental trials were performed to evaluate the molecular solubility at different SCX4 concentrations, identifying 15mg/ml SCX4 in water as the saturation limit (at this concentration, the solution does not become completely clear after 24 h). The solutions were processed by using a built-up electrospin-ning machine equipped with a high voltage (Gamma High Voltage, Ormond Beach, FL, USA) working at 13 kV and a syringe pump (KD Scientific, Holliston, MA, USA) with flow rate set to 0.8 mL/h. The distance between the ground collector and the needle (20 G) was fixed to 100 mm. Electrospun fibres were fabricated under standard environmental condi-tions – i.e., temperature 28ï‚°C and relative humidity of 50% ± 5% - and, after the deposition, dried at room temperature to remove any residual solvents. All the samples were stored in a desiccator for the further characterization”.
2) Several sentences in the INTRODUCTION about the developments of electrospinning should be included for provoking the readers’ interests (e.g. SCX is loaded in the core of core-shell nanofibers or one side of Janus nanofibers), and in turn the impact of your article after publication, e.g. electrospinning is fast developing from the single-fluid bledning processes (https://doi.org/10.3390/pharmaceutics14071494;Pharmaceutics, 4(3),484; Pharmaceutics, 4(4),777) to coaxial (DOI10.3390/pharmaceutics14071442; ), triaxial (https://wires.onlinelibrary.wiley.com/doi/full/10.1002/wnan.1772), side-by-side (https://doi.org/10.1016/j.bioadv.2022.212795), tri-layer side-by-side processes (Doi: 10.1007/s42114-022-00478-3) for creating nanostructures. However, these structures are similar as the monolithic nanofibers that active ingredients must be loaded into them to endow the designed functional performances.
Thank you for your comment. The introduction section has been improved and a sentence has been included also in the conclusion section to underline the perspective use of the electrospinning in combination with supramolecular species like calixerines. Accordingly, three new references have been included, as you suggested (refs 58-60):
“58. Hauck, M.; Dittmann, J.; Zeller-Plumhoff, B.; Madurawala, R.; Hellmold, D.; Kubelt C.; Synowitz M.; Held-Feindt J.; Adelung R.; Wulfinghoff S.; Schütt F. Fabrication and Modelling of a Reservoir-Based Drug Delivery System for Customizable Release. Pharmaceutics 2022, 14(4), 777; https://doi.org/10.3390/pharmaceutics14040777
59.Kazsoki A.; Palcsó B.; Omer S.M.; Kovacs Z.; Zelkó R. Formulation of Levocetirizine-Loaded Core-Shell Type Nanofibrous Orally Dissolving Webs as a Potential Alternative for Immediate Release Dosage Forms. Pharmaceutics, 2022 Jul 11;14(7):1442. doi: 10.3390/pharmaceutics14071442
60.Xua, H.; Zhang, F.; Wang, M.; Lv, H.; Yua, D-G., Liu, X.; Shen H. Electrospun hierarchical structural films for effective wound healing. Biomat Adv, 136, 2022, 212795.”
3) What is the purity of HFIP? How about the saturated solubility of SCX4 in it?
HFIP (Sigma Aldrich) purity is 99%. HFIP is a polar solvent and miscible with SCX4 for high concentrations as reported in literature. In our case, saturation limit is over the SCX4 concentration (10/15mg in 10 ml) used in this work. A sentence was included into Materials and Methods section to clarify this aspect.
4) The scale bars in Figure 3 please be enlarged.
Thank you for your comment. SEM images were taken with SEM JEOL 6360 LA, Japan, and scale bar was automatically burn onto the image. To improve the readability scale bar of the images was reported also into the caption text.
5) Line 291, Water contact angle.
Thank you, the text has been revised.
6)The references’ formats please be unified, and the most recent three years’ references can be more, a common standard is 25%, particularly for a fast developing region. PHARMACEUTICS has many excellent publications about electrospinning. The authors are suggested to discuss them with your job.
Four new and pertinent references have been included. Among them, two from Pharmaceutical MDPI journal.
Reviewer 3 Report
The article reports promising data concerning the fabrication of nanostructural innovative platforms that may potentially be used for different therapeutic applications – from detection to controlled drug delivery. Just two issues to be addressed:
1. Figure 3 should be placed after (not before) its first mention in the text. Also, the scale bar is hardly visible, and should be improved.
2. The explanation why the L929 cell line was used for cytotoxic study will improve the article.
Author Response
The article reports promising data concerning the fabrication of nanostructural innovative platforms that may potentially be used for different therapeutic applications – from detection to controlled drug delivery. Just two issues to be addressed:
1) Figure 3 should be placed after (not before) its first mention in the text. Also, the scale bar is hardly visible, and should be improved.
Figure 3 had been revised. SEM images were taken with SEM JEOL 6360 LA, Japan, that automatically burn the scale bar onto the image. In order to improve the readability, scale bar has been indicated also into the caption text.
2) The explanation why the L929 cell line was used for cytotoxic study will improve the article.
Thank you for this question. L929 is a normal fibroblast cell line from subcutaneous connective tissue of mouse and are considered one of the most reliable cell model to investigate cytotoxicity. In this study, we proposed to use them also because, after several passages, these fibroblasts transform into cells that have features of cancerous cells. This makes really interesting to design an in vitro model for cancer therapy, to investigate the release of chemotherapeutic drugs via calixarenes. A sentence has been included into the manuscript, in the results and discussion section.
Round 2
Reviewer 1 Report
The author has revised well in the current version.
Reviewer 2 Report
In general, the authors have improved the manuscript’s quality substantially. Although there are some tiny inappropriate places in the texts and also the references’ formats, they can be corrected during publication.
I recommend its acceptance for publication in its present form.
Reviewer 3 Report
The article reports the results of electrospun nanofibres design with sulfonatocalix[4]arene for biomedical application. Introduction of ulfonatocalix[4]arene into nanofibers resulted in improved hydrophilic properties. Sufficient biocompatibility with normal cell culture was shown.